# Influence of Electrospun Fibre Secondary Morphology on Antibiotic Release Kinetic and Its Impact on Antimicrobic Efficacy

**DOI:** 10.3390/ijms241512108

**Published:** 2023-07-28

**Authors:** Mariella Rosalia, Pietro Grisoli, Rossella Dorati, Enrica Chiesa, Silvia Pisani, Giovanna Bruni, Ida Genta, Bice Conti

**Affiliations:** 1Department of Drug Sciences, Pharmaceutical Section, University of Pavia, Via Taramelli 12, 27100 Pavia, Italy; mariella.rosalia01@universitadipavia.it (M.R.); rossella.dorati@unipv.it (R.D.); enrica.chiesa@unipv.it (E.C.); silvia.pisani@unipv.it (S.P.); ida.genta@unipv.it (I.G.); 2Department of Drug Sciences, Pharmacological Section, University of Pavia, Via Taramelli 16, 27100 Pavia, Italy; 3Consorzio per lo Sviluppo dei Sistemi a Grande Interfase (C.S.G.I.), Department of Chemistry, Physical Chemistry Section, University of Pavia, Via Taramelli 10, 27100 Pavia, Italy; giovanna.bruni@unipv.it

**Keywords:** vascular graft infection, small-diameter vascular graft, tobramycin, secondary fibre morphology, drug release kinetic, drug release modelling, antimicrobial activity

## Abstract

Vascular graft infections are a severe complication in vascular surgery, with a high morbidity and mortality. Prevention and treatment involve the use of antibiotic- or antiseptic-impregnated artificial vascular grafts, but currently, there are no commercially available infection-proof small-diameter vascular grafts (SDVGs). In this work we investigated the antimicrobic activity of two SDVGs prototypes loaded with tobramycin and produced via the electrospinning of drug-doped PLGA (polylactide-co-glycolide) solutions. Differences in rheological and conductivity properties of the polymer solutions resulted in non-identical fibre morphology that deeply influenced the hydration profile and consequently the in vitro cumulative drug release, which was investigated by using a spectrofluorimetric technique. Using DDSolver Excel add-in, modelling of the drug release kinetic was performed to evaluate the release mechanism involved: Prototype 1 showed a sustained and diffusive driven drug release, which allowed for the complete elution of tobramycin within 2 weeks, whereas Prototype 2 resulted in a more extended drug release controlled by both diffusion and matrix relaxation. Time-kill assays performed on *S. aureus* and *E. coli* highlighted the influence of burst drug release on the decay rate of bacterial populations, with Prototype 1 being more efficient on both microorganisms. Nevertheless, both prototypes showed good antimicrobic activity over the 5 days of in vitro testing.

## 1. Introduction

Vascular grafts (VGs) are biological or artificial devices used as replacements for damaged blood vessels, especially arteries, to treat several vascular diseases, including coronary arterial disease (CAD), peripheral arterial disease (PAD) and aneurysmal degeneration of the thoracic or abdominal aorta [1]. Based on their origin, VGs are classified into autologous grafts, when vascular tissue from the same patients is used; allografts, when tissue from a donor is implanted; xenografts, when animal vascular tissue derived from bovine or swine is employed; or synthetic grafts, when they are composed of artificial biomaterials [2]. Due to the growing incidence of cardiovascular diseases [3] and therefore also to the request of VGs, particular interest towards the development of synthetic vascular grafts has increased. Currently, commercially available synthetic VGs are predominantly made of non-degradable polyethyleneterephthalate (PET, Dacron) or expanded polytetrafluoroethylene (ePTFE), which are used to produce woven/knitted or nontextile grafts, respectively [4]. These materials are suitable for large-diameter vessel surgery but lack the adequate properties for the replacement of small-calibre vessels with a diameter equal to, or smaller, than 6 mm. In fact, PET and ePTFE present scarce endothelisation of the graft’s lumen, leading to reduced patency and thrombosis, as well as mechanical mismatch, with the native tissue causing intimal hyperplasia [5]. Moreover, non-degradable biomaterials induce an intense and chronic inflammatory reaction in the implant site: the VG is recognised as a foreign body by the immune system, which then attempts to destroy and/or isolate it from the surrounding tissue [6]. The activity of extracellular enzymes and reactive oxygen species can lead to the erosion of the biomaterial and loss of mechanical strength, with a high risk of dilatation and rupture of the graft [6,7]. Synthetic vascular grafts are also characterised by a higher risk of postsurgical infection when compared to auto- and allografts. Therefore, they are still being used only as alternative option in vascular surgery. Since vascular diseases can compromise the overall health of the patient’s blood vessels, the harvesting of small-diameter autografts could be impossible or lead to severe co-morbidity. Moreover, the scarce availability of allografts and the high risk of immune rejection in the use of xenografts poses important limitations in the use of such grafts, leading to a major reduction in the available treatment choices [4,5].

Vascular graft infections (VGIs) have a low incidence of 1 to 6% but represents a severe and life-threatening complication in vascular surgery. Incidence depends on the anatomical localisation of the VG: intracavitary grafts are less prone to infection, but aortic VGIs especially have a mortality between 25 and 75%; and extracavitary grafts, e.g., in the limbs, have a higher risk of infection and correlated morbidity, such as amputation (40%) but with a lower mortality (17%) [8]. VGIs are particularly challenging to diagnose because of the aspecificity of the first symptoms (fewer, inflammation and pain of the surgical wound, limb ischemia), difficulty in reaching the anatomical site, the possibility of false-positive imaging and the possible latency of occurrence (even 6 to 12 months after surgery) [9]. The microorganisms involved in early VGIs (onset within 3–4 months) are Gram-positive Staphylococci, including *Staphylococcus aureus* (75% of infections) and methicillin-resistant *S. aureus*; and Gram-negative microorganisms, such as *Escherichia coli*, *Pseudomonas aeruginosa* and *Klebsiella* species, are also involved. Early infections are more likely to cause graft rupture and pseudoaneurysm. In late infections (onset 3–4+ months after surgery), the causative Gram-positive microorganisms are *Staphylococcus epidermidis*, and other bacteria of the skin flora, such as *Propionibacterium acne* and *Corynebacterium* sp., while the Gram-negative microorganisms include *Enterobacter*, *Serratia* and *Proteus* species. Moreover, polymicrobial infections with biofilm formation are common, especially in late VGIs, increasing the risk of reinfection after treatment [4,8]. VGIs generally occur after intraoperative or postoperative contamination of the graft and surgical wound. A higher infection risk has to be expected in emergency surgeries, prolonged operating time, lymphatic manipulation, poor wound healing and skin necrosis. Grafts localised in the groin or gut area are more prone to infection. Moreover, co-morbidity and high patient age increase the risk of VGIs [8]. Two main strategies are employed for the treatment of VGIs, namely antibiotic administration and surgical intervention. When an infection is suspected, empirical treatment with intravenous antibiotics against the most-likely causative microorganism is immediately performed, generally using wide-spectrum antibacterial drugs. The treatment is then adapted after microbiological analyses of blood cultures and intraoperative sampling. The duration of the treatment can be variable and depends on the onset of the infection (early or late), the severity and the outcome of the revision vascular surgery. Intravenous antibiotics are usually administered for a minimum of 2 weeks and up to 6 weeks, followed by oral administration of up to 4 weeks. Surgical intervention has to be performed as soon as possible and consists of the removal of the infected graft, aggressive wound debridement and revascularisation with a new graft. Depending on the localisation of the infection, severity and patient conditions, the surgeon can also decide for a more conservative strategy and perform a partial excision of the graft. In some cases, if the graft cannot be removed or if biofilm formation has led to chronic infection, life-long antibiotic treatment is necessary, exposing the patient to an increasing risk of developing antibiotic resistance and leading to high exposure to drug side effects [9,10]. 

Due to the severity of VGIs, prevention strategies had been proposed, including pre-operative antibiotic treatment and the use of antibiotic- or antiseptic-impregnated vascular grafts. Currently, few antimicrobial VGs are commercially available, including elemental or acetate-silver-coated/impregnated PET grafts, or rifampicin-bonded or -soaked PET grafts. The latter could be with or without the synergistic effect of silver and triclosan [11,12]. These grafts have proven to be effective in aorta or lower limb vascular surgery but are not available in a calibre compatible with small-diameter arteries. Moreover, the antimicrobial agent release from such types of grafts is generally very fast and therefore not suitable for a prolonged antibiotic therapy [10]. A more innovative approach consists of the use of nanotechnology for the development of tissue-engineering vascular grafts (TEVGs) with antimicrobial properties. TEVGs are tubular scaffolds intended as support for cell attachment and proliferation, with the final goal of promoting tissue regeneration. Upon implantation, when biodegradable materials are used for the manufacture of TEVGs, the slow breakdown of the scaffold allows for the ingrowth of new tissue and the substitution of the biomaterial with the extracellular matrix (ECM) [13]. Moreover, nanotechnologies can be used for the development of drug delivery systems with peculiar properties thanks to the nanosized carriers’ high volume-to-surface ratio; for example, antibiotic-loaded nanoparticles have been shown to have improved activity on biofilms and to overcome antibiotic resistance [14]. 

With this premises, there is an urgent need for the development of synthetic small-diameter vascular grafts suitable for tissue engineering purposes and resistant to infection, to fill the existing therapeutic void and improve the prognosis of vascular surgery patients. The aim of this work is to investigate the influence of the morphology of nanostructured small-diameter vascular grafts on the release of embedded tobramycin and to evaluate in vitro the antimicrobic efficacy of the drug release profile on two representative bacterial strains, *Staphylococcus aures* and *Escherichia coli*. Two small-diameter vascular graft prototypes (PR1_T and PR2_T) were produced by electrospinning tobramycin-loaded polylactide-co-glycolide (PLGA) solutions. An evaluation and critical discussion was carried out from the physical-chemical and biological stand points.

## 2. Results

### 2.1. Polymer Solutions Characterization

Rheological and conductivity properties were studied to determine their influence on the primary and secondary morphology of small-diameter vascular graft prototypes. Rheological characterization of 10% and 12.5% *w*/*v* PLGA solutions showed significant differences in linear viscoelastic regions (LVERs) and zero shear rate viscosities among the two polymer concentrations. LVERs of 10% *w*/*v* PLGA placebo and 0.2% *w*/*w_PLGA_* tobramycin-doped polymer solutions had the same range of shear strain, between 1% and 25% (Figure 1a,b). This shear strain range corresponds to a shear stress range of 0.039 ± 0.001 Pa and 1.187 ± 0.055 Pa for the placebo polymer solution, and 0.038 ± 0.003 Pa and 1.123 ± 0.036 Pa for the tobramycin-loaded solution. More concentrated 12.5% *w*/*v* placebo and tobramycin-doped polymer solutions were characterised by a wider LVER range (Figure 1c,d) from 0.099% to 40%. This indicated polymer solution stability in a wider range of shear stresses, ranging from 0.0096 ± 0.0008 Pa to 6.021 ± 0.184 Pa for the placebo PLGA solution, and 0.0096 ± 0.0009 Pa to 6.163 ± 0.015 Pa for the tobramycin-doped polymer solution. Hence, for both polymer solutions, tobramycin addition did not change LVERs and maximum bore shear stress, which were indeed dependent on polymer concentration. Moreover, polymer concentration significantly affected the zero-shear rate viscosity η_0_, with it being lower and equal to 0.694 ± 0.010 Pa × s for the 10% *w*/*v* polymer solution and 2.142 ± 0.205 Pa × s for the 12.5% *w*/*v* polymer solution. Even in this case, no significant differences were reported after the addition of 0.2% *w*/*w_PLGA_* tobramycin to the PLGA solution, having zero-shear rate viscosities of 0.701 ± 0.033 Pa × s and 2.233 ± 0.203 Pa × s for the 10% and 12.5% *w*/*v* tobramycin-doped PLGA solutions, respectively (Figure 2a). 

The loading of tobramycin into polymer solutions significantly reduced the conductivity of the polymer solutions: 10% *w*/*v* placebo and 0.2 *w*/*w_PLGA_* tobramycin-loaded PLGA solutions had a conductivity of 9.697 ± 0.888 μS/cm and 5.086 ± 0.466 μS/cm, respectively, while the conductivity of 12.5% *w*/*v* placebo and 0.2 *w*/*w_PLGA_* tobramycin-loaded polymer solutions were 9.417 ± 0.979 μS/cm and 5.086 ± 2.049 μS/cm for, respectively (Figure 2b). No significant differences were reported among placebo solutions with different polymer concentrations and between drug-doped solutions with different polymer concentrations, hence the PLGA percentage of the solution does not influence the conductivity.

The theoretical shear stress was calculated based on the polymer solutions’ zero-shear viscosity, needle internal diameter and flow rate used to electrospin the polymer solutions (Table 1). The 10% *w*/*v* polymer solution was subjected to 1.264 ± 0.008 Pa shear stress, whereas 12.5% *w*/*v* polymer solution was subjected to 11.89 ± 0.35 Pa. According to the aforementioned maximum shear stress values of the polymer solution in LVER, 12.5% *w*/*v* polymer solutions were submitted to significantly higher (*p* < 0.01) shear stress, whereas 10% *w*/*v* solutions were subjected to a shear force similar to the maximum supported value in the LVER.

### 2.2. Morphological Characterization of Small-Diameter Vascular Grafts Prototypes

Prototype 1 (PR1) vascular grafts were produced by electrospinning 12.5% *w*/*v* PLGA solution, whereas Prototype 2 (PR2) grafts were made with 10% *w*/*v* polymer solution. To obtain tobramycin-loaded vascular grafts (PR1_T and PR2_T) the polymer solutions were doped with 0.2% *w*/*w_PLGA_* of antibiotics prior to electrospinning. Electrospinning parameters were optimised and tailored according to polymer solution characteristic and are reported in Table 1. The morphological properties of PR1, both placebo (PR1_PL) and tobramycin-loaded (PR1_T) grafts, were preliminary investigated in a previous work of the same authors [15]. Here, the investigation is focused on the comparison between placebo and drug-loaded PR2 grafts (PR2_PL and PR2_T, respectively). Therefore, the overall fibre and pore morphology are reported, as they are more relevant for the scope of this work. 

Scanning electron microscopy images of PR1 and PR2 vascular grafts at different magnifications are displayed in Figure 3. For both prototypes, well-formed and randomly distributed fibres were observed. PR1 secondary fibre morphology was characterised by net-like cracks on the surface of single fibres that were absent in PR2, which showed a uniform surface with a slight prickle-like rugosity. As disclosed by the data reported in Table 2, PR1 had significantly (*p* < 0.01) thicker graft walls with respect to PR2, whereas the loading of tobramycin did not affect the prototypes’ thickness. PR1 also displayed significantly (*p* < 0.001) bigger fibres in the micrometre range, whereas PR2 had a fibre diameter in the nanometre range, between 0.794 μm and 0.870 μm. In PR1_T grafts, the presence of tobramycin led to a reduction in fibre diameter (*p* < 0.001) and a narrowing of fibre-size distribution (Figure 4a). On the contrary, in PR2_T, the drug led to an increase (*p* < 0.01) of fibre diameter and to a less uniform size distribution (Figure 4a). Even for pore size, PR1 displayed higher pore diameter of between 8 and 39 μm, while PR2 had smaller pores with a size between 2.7 and 19 μm (Figure 4b). Moreover, in both prototypes, the loading of tobramycin did not affect pore size and their diameter distribution and was therefore not statistically significant (*p* > 0.05). The average porosity did not change among the four types of grafts and ranged between 43.79% and 45.85%. 

### 2.3. Small-Diameter Vascular Graft Prototypes Hydration Profiles

Hydration profiles of PR1_T and PR2_T in PBS pH = 7.4 and at 37 °C were performed to better understand the grafts’ drug release mechanism and are displayed in Figure 5. The two prototypes showed different water uptake rates and overall hydration capacity. PR1_T reached the steady state after 48 h of incubation and was able to uptake only 2.5 times of its starting weight. On the contrary, PR2_T was able to reach 400% water uptake but needed 216 h (9 days) to reach maximum hydration. 

### 2.4. In Vitro Drug Release Profiles and Kinetics of Small-Diameter Vascular Graft Prototypes

The extraction of encapsulated tobramycin into electrospun vascular grafts prototypes was performed, and the quantification of the drug was evaluated using a spectrofluorimetric method. PR1_T and PR2_T resulted having significantly different (*p* < 0.05) encapsulation efficiencies of 45.95 ± 6.52% and 35.19 ± 2.85%, respectively. Hence, a different (*p* < 0.05) drug content was also determined for PR1_T and PR2-T, being equal to 0.92 ± 0.11 μm/mg and 0.70 ± 0.05 μm/mg, respectively.

The tobramycin dissolution profile is shown in Figure 6b: immediately after adding dissolution medium, more than 50% of the drug was dissolved, reaching 100% of dissolution after 9 h. If compared to the cumulative tobramycin release from both prototypes in the first week of in vitro drug release assay, extended release of the drug was ensured by both vascular graft prototype. The entire cumulative tobramycin release PR1_T and PR2_T for a time period of up to 840 h (35 days) is displayed in Figure 6a. PR1_T and PR2_T both showed a burst release of tobramycin within the first 9 h without lag-phase. The PR1_T burst release was significantly (*p* < 0.01) higher than that of PR2_T, reaching a mean drug release of 65.0 ± 8.0% after 24 h, whereas PR2_T reached only 24.2 ± 8.5% of released tobramycin. PR1_T approached 100% drug release only after 336 h (14 days), whereas PR1_T did not complete drug release during the in vitro test, reaching a maximum of 77.4 ± 21.0% at 840 h (35 days). Because of the high variability of PR2_T release data, after 96 h (4 days), statistical differences between the two prototypes were no longer detected. 

### 2.5. Drug Release Kinetic Evaluation for Vasculas Graft Prototypes

Cumulative percentage drug release data of PR1_T and PR2_T were fitted to seven different equations commonly used to describe drug release kinetics, including Higuchi, Korsmeyer–Peppas, Peppas–Sahlin, Weibull, Logistic, Gompertz and Probit equations. The goodness of fit of the different models were estimated through several statistical criteria, of which the values for each studied mathematical model are listed in Table 3. 

Among the seven different models fitted to Prototype 1 drug release data, the best match was reported for Weibull model, presenting a high correlation (R^2^ = 0.9978) between the experimental data and predicted data. A high corrected correlation parameter value (R^2-adj^ = 0.9891), low AIC (108) and MSC higher than 3 (4.1387) indicated excellent model fitting. The Higuchi model was discarded for having a negative correlation and MSC parameters. The Gompertz and Logistic models displayed comparable AIC and MSC to Weibull model, but R^2^ and R^2-adj^ had lower values. The Probit model showed to have even lover AIC and higher MSC than the Weibull model, indicating good model fitting, but R^2^ and R^2-adj^ displayed lower values. Prototype 2 showed excellent fitting with the Peppas–Sahlin model, having very high correlation parameters values (R^2^ = 0.9909 and R^2-adj^ = 0.9905), the lowest AIC (100) and the highest MSC (4.5131). A comparable fitting was shown by the Korsmeyer–Peppas model. Intermediate correlation parameters values were reported for Higuchi, Weibull and Probit models, having also high AIC values and MSCs lower than 3. The worst fitting was shown by the Gompertz model, with R^2^ and R^2-adj^ lower than 0.9, an AIC of 154 and a MSC lower than 2.

Table 4 reports the equation of the best fitting models for each vascular graft prototype and the calculated equation parameter values. As already mentioned, Prototype 1 release was best described by the empirical Weibull model, as shown also by the superimposed real and predicted cumulative drug release profiles in Figure 7a. Weibull’s α parameter is also called the scale parameter and defines the timescale of the drug release process; β is the form parameter describing the form of the curve, with β > 1 meaning that the drug release profile had a parabolic shape; Ti is the localisation parameter indicating the latency time of drug release. Prototype 2 is instead best described by the Peppas–Sahlin equation, a mechanistic model, as shown in Figure 7b. Parameter k_1_ is the constant indicating the rate of diffusional drug release, k_2_ is the constant describing the rate of relaxation of a matrix drug release system, and exponent *m* is the parameter describing the Fickian diffusion for a drug release system of any shape.

### 2.6. Antimicrobic Activity of Small-Diameter Vascular Graft Prototypes

The antimicrobic activity of PR1_T and PR2_T was studied against *Staphylococcus aureus* and *Escherichia coli*. In a previous work the authors reported the results of this assay on PR1_T [15], showing a significant antibacterial efficacy only after 24 h of contact. The microbicidal efficiency (ME) then stabilised at 8-log reductions after 48 h for *E. coli* and after 72 h for *S. aureus*, lasting until the end of the test (120 h). PR2_T antimicrobic efficacy is reported in Table 5. As already observed for PR1_T, *E. coli* had a faster population decay rate than *S. aureus*, with an ME equal to 8 only after 48 h and therefore a comparable efficacy to PR1_T. *S. aureus* showed a slower microbicidal rate, reaching an ME close to 8 only after 96 h. Still, PR2_T was able to completely break down the starting bacterial load of 10^7^ CFU/mL within the 5 days of testing. 

## 3. Discussion

Tissue-engineered vascular grafts (TEVGs) are tubular prostheses produced with biomaterials that support the attachment and proliferation of cells, specifically endothelial cells (ECs), smooth muscular cells (SMCs) and fibroblasts (FBs) and promote vascular tissue remodelling and regeneration. To fulfil this task, the selection of appropriate biomaterials and manufacturing techniques is fundamental [16]. Natural polymers, such as collagen, gelatine, elastin, fibronectin, lecithin and silk fibroin, are components of the extracellular matrix (ECM) or present molecular cues that resemble the ECM, providing adequate cell attachment and cell signalling for cell expansion, ensuring good biocompatibility. However, such biomaterials are characterised by inadequate mechanical properties, rapid degradation, and limited processability. Moreover, the production and/or extraction is expensive and characterised by a high batch-to-batch variability [17]. For these reasons, synthetic biopolymers are extensively used for the manufacturing of tissue-engineering scaffolds. In particular, aliphatic polyesters, such as polylactic acid, polyglycolic acid, polycaprolactone, polylactic-co-glycolic acid, polylactic-co-caprolactone and polyglycerol sebacate, were extensively studied for their application in vascular regenerative medicine. They are produced in a reproducible and cost-effective manner and are characterised by good and tunable processability, mechanical properties and biocompatibility. Furthermore, aliphatic polyesters can be biodegraded through hydrolytic or enzymatic pathways, yielding non-toxic products that can be easily metabolised and eliminated by the human body [18]. Their main drawback is the absence of ECM-like molecular cues that impair cell attachment and proliferation, but through specific surface treatments or bonding to functional groups, physical or chemical modifications that enhance cell–biomaterial interactions can be achieved [19]. 

In this work, polylactic-co-glycolic acid was the selected biomaterial for the production of small-diameter vascular graft prototypes. It is one of the most used polyesters in tissue engineering and drug delivery. PLGA is FDA approved and is commercially available at medical grade and at several molecular weights and PLA:PGA ratios, having therefore tunable biodegradation rates. Its good processability is compatible with different nanostructured-scaffold production techniques, including electrospinning [20]. PLGA was studied for vascular tissue engineering applications, resulting in suitability for endothelial and smooth vascular cell attachment and expansion, alone or in combination with other polymers [21,22,23,24,25]. The electrospinning technique was selected because of the possibility to obtain tubular fibrous scaffolds in a straightforward one-step procedure. Electrospun fibres are known for resembling the nanofibrous ECM structure and are therefore widely used in the tissue engineering field [26]. If compared to polymer films, electrospun fibres also seem to induce the formation of thinner fibrous capsules, reducing scaffold isolation [27,28]. 

PLGA polymer was solubilised in dichloromethane (DCM), a low-boiling solvent that ensures optimal polymer dissolution and the formation and drying of electrospun fibres. Dimethylformamide (DMF) was added as a cosolvent due to its high dielectric constant, promoting fibre stretching and thinning during electrospinning. A solvent system ratio of 75:23.5 DCM/DMF was optimised in a previous work [15]. The electrospinning procedure is based on the application of an electric field to a polymer solution that is extruded through a thin needle: the applied electrical potential causes the deformation of a polymer solution drop to a conical shape. The deformation is caused by the accumulation and consequent repelling of electric charges on the surface of the polymer solution. When the applied voltage exceeds a critical value, the intensity of the electrical repulsion overcomes the surface tension of the polymer solution, forming a Taylor cone from which a polymer jet departs. The electrical charges in the jet cause its whipping, leading to fibre stretching and thinning. The obtained fibres are finally collected on a grounded collector. Several parameters can influence the morphology of the obtained fibrous scaffold, including the process parameters (applied voltage, polymer solution flow rate, needle internal diameter, needle-to-collector distance, collecting electrode), solution parameters (polymer concentration and molecular weight, used solvents, solution viscosity, conductivity, surface tension), and ambient parameters (temperature, relative humidity). To evaluate the effect of solution parameters on the morphology of the obtained electrospun fibres, rheological and conductivity measurements were performed on polymer solutions. PLGA concentrations of 12.5% and 10% *w*/*v* were used to produce Prototype 1 (PR1_PL) and Prototype 2 (PR2_PL) small-diameter vascular grafts, respectively. To obtain tobramycin-loaded vascular grafts (PR1_T and PR2_T), 0.2% *w*/*w_PLGA_* was added directly into the polymer solutions. Rheological analyses highlighted that the addition of this amount of drug did not influence the viscoelastic properties and the zero-shear viscosity of the polymer solutions. As expected, these two parameters were dependent on the polymer concentration: 12.5% *w*/*v* PLGA solutions had a wider linear viscoelastic region (Figure 1c,d), and therefore are stable under a wider range of shear stresses; moreover, 12.5% *w*/*v* polymer solution also had a three-times higher viscosity than the 10% *w*/*v* PLGA solution (Figure 2a). In fact, a higher polymer concentration is related to an increase in the number of polymer molecules and of cohesive chain–chain interactions, which reflects on the flowing properties (viscosity) and stability under shear stress (LVER) of the polymer solution [29]. On the other hand, polymer solution conductivity was not affected by polymer concentration, since PLGA does not have a chemical structure suggesting strong electrical conductivity [30]. Solution conductivity was mostly dependent on the use of DMF in the solvent system, which has a high dielectric constant (ε = 35.51 at 35 °C). Interestingly, the polymer solutions conductivity was rather influenced by tobramycin loading, which significantly lowered it (Figure 2b). While tobramycin is highly soluble in aqueous solvents, inorganic DCM and DMF are non-solvents, causing immediate precipitation of the drug and leading to the formation of spherical particles between 10 and 60 μm in diameter. The so-obtained particle suspension is stabilised by the high viscosity of the PLGA solution limiting particle aggregation and caking; nevertheless, electric interaction between suspended particles is still possible. Tobramycin is characterised by five amino groups that can be ionised and positively charged. In a suspension, the electrophoretic mobility of the particles is responsible for the suspension’s conductivity. This conductivity is directly proportional to particle concentration, but inversely dependent on particle size because of the reduced surface area [31]. Hence, the reduction in conductivity of both 10% and 12.5% *w*/*v* PLGA solution conductivity could be ascribed to the presence of big, suspended particles with reduced electrophoretic mobility, acting as obstacles to the movement of charges throughout the polymer solutions.

Polymer solution rheological and conductivity properties influenced electrospun fibres morphology. PR1_PL and PR1_T polymer solutions gained overall bigger fibres in the micrometre range and also bigger pores, while PR2_PL and PR2_T fibres were in the nanometre range and small pores were obtained (Table 2). This difference can be ascribed to both the viscosity and process parameters: higher viscosities values lead to less stretching of the polymer jet due to more intense polymer chain interactions. Moreover, an increased polymer flow rate results in a higher amount of polymer solution in the polymer jet, resulting in bigger fibres and also bigger pores formation [26]. Indeed, the combination of higher viscosity (Figure 2a) and higher flow rate (Table 1) used to electrospin 12.5% *w*/*v* PLGA solutions increased the fibre and pore diameter. On the other hand, the reduction in polymer solution conductivity due to tobramycin addition to the polymer solutions seemed to have contrasting effects. For PR2_T, a significant increase in fibre diameter and also scattering of fibre distribution was observed, whereas in PR2_PL smaller and homogeneous fibres were obtained. This is because the higher polymer solution conductivity increased the stretching of the polymer solution jet [26]. The opposite trend was observed for PR1: less conductive tobramycin-loaded solution gained small and homogeneous fibres. This behaviour might be explained by considering the shear stress applied to the polymer solution during extrusion from the needle. As calculated by applying Equations (1) and (2), the 12.5% *w*/*v* polymer solution was subjected to an extrusion shear stress two-times greater than the maximum stress of the LVER, leading to an inhomogeneous polymer solution flow due to the partial disruption of the viscoelastic system. This phenomenon could be enhanced by the higher conductivity that exerted a pulling action on the extruded PLGA solution and the combination of which resulted in the thickening of a part of the electrospun fibres. In fact, it is well known that poorly entangled polymer solutions and high flow rates are related to the formation of beaded or thickened fibres [32]. 

Another important difference between PR1 and PR2 regards the secondary fibre morphology: the presence of net-like cracks on PR1_PL and PR1_T fibres surface was observed (Figure 3c,f). Porous electrospun nanofibers can be obtained by using phase separation techniques: volatile solvents, such as DCM, chloroform, tetrahydrofuran or acetone are used in combination with high-boiling solvents, including DMF, cyclohexanone or N,N-dimethylacetamide, to induce phase separation in the polymer jets during electrospinning [33]. In non-solvent-induced phase separation (NIPS), the presence of a slow evaporating non-solvent miscible with the main volatile polymer solvent is responsible for the formation of solvent pockets during the rapid evaporation of the volatile solvent. This leads to the formation of round ad elliptical pores elongated in the longitudinal direction of the fibre [34]. But since the cracks on the surface of the PR1 fibres were not rounded but were rather net-like and directed in both longitudinal and circumferential direction, NIPS or other phase separation techniques cannot be applied to explain the secondary morphology of these fibres. We hypothesise that the destabilizing effect of high shear stress on the 12.5% *w*/*v* polymer solution also had an impact on secondary fibre morphology: the visibly cracked surface on both PR1_PL and PR1_T is probably due to discontinuous polymer solution flow. Indeed, PR2_PL and PR2_T fibres showed a continuous fibre surface because they did not experienced disruptions of the viscoelastic properties of the polymer solution since the applied extrusion shear stress was compatible with the LVER. Moreover, in PR2_PL and PR2_T fibres, a so-called cactus or prickled surface was observed (Figure 3i,l): this morphology is ascribed first to the formation of a continuous, dense, and mechanically strong polymer layer on the surface of the fibre due to the rapid evaporation of a volatile solvent. A small fraction of non-solvent mixed with volatile solvent is still trapped in the core of the fibres and escapes from the fibre by erupting at the weaker spots of the external polymer layer of the fibre. At the eruption points the polymer immediately precipitates, forming the characteristic prickles [35]. To the best of our knowledge, this is the first time that this secondary fibre morphology is described for PLGA fibres.

The two profoundly different electrospun primary and secondary fibre morphologies in PR1_T and PR2_T had a fundamental impact on the hydration and drug release profiles of the two prototypes. The PR1_T hydration process was faster, reaching a plateau value after 48 h of immersion. On the other hand, PR2_T fluid uptake was slower, with a plateau value reached after 216 h (9 days) of testing. In addition, PR2_T showed an overall higher water uptake percentage of about 400% against the 250% equilibrium water uptake of PR1_T (Figure 5). Based on the discussion above, we hypothesise these differences are strictly related to the fibre and pore morphology of the two scaffolds: the bigger scaffold’s pores and especially the net-like cracks of the PR1_T fibres surface were responsible for an enhanced hydration rate. On the other hand, PR2_T smaller scaffold’s pores and continuous fibre surface reduced the hydration rate; the smaller fibres increased the surface area of the scaffold, increasing the amount of water retained by the prototype.

Since tobramycin is a highly water-soluble drug (Figure 6b), the hydration rate influenced the drug release. The drug release assay was performed in phosphate-buffered saline (PBS). PBS is an electrolyte solution with composition, osmolarity and pH close to extracellular fluid, and it is well known for its very good stability even for longer time periods. It is therefore the most used medium for the study of drug release kinetics. The drug content of PR1_T and PR2_T small diameter vascular grafts was different, hence the differences in drug release profiles might be ascribed to the reservoir/drug gradient effect in PR1_T. Nevertheless, this hypothesis is not sufficient to justify the prototypes’ drug release kinetics. Because PR1_T was completely hydrated after 48 h, the encapsulated drug was dissolved at a fast rate, inducing an immediate burst release of tobramycin. This led to the leakage of more than half of the drug within the first 24 h, and drug release was complete after 2 weeks (Figure 6a). Kinetic modelling of the PR1_T drug release profile showed that the Weibull model had the highest predictive power. The correlation parameter R^2^ was 0.9978, indicating very good matching experimental data and predicted data; by using the corrected correlation parameter R^2-adj^, comparisons between different models with different numbers of variables was possible, and a R^2-adj^ of 0.9891 indicated a good prediction power (Table 3). The Akaike information criterion (AIC) and its modified reciprocal form, the model selection criterion (MSC), are two selection criterions used to estimate the predictive error of the model, considering both over- and underfitting. They allow for the evaluation of the quality of the model when more variables are involved and the selection of the best fitting model among a library. The difference between AIC and MSC is that the former is dependent on the magnitude of the data and on number of data points, whereas the latter is normalised to make it independent from the scaling of data points. There is no threshold value to determine the best fitting model when AIC is used, but when a comparison between different models is made, a low AIC indicates good prediction power. For MSC the trend is opposite, and higher values are correlated to good fitting; generally, MSC higher than 3 indicates good prediction power [36,37]. Hence, the Weibull model was chosen among the seven tested models also because of its low AIC (108) and high MSC (4.1387) values (Table 3). The Weibull model is an empirical model that describes the linear relationship between the logarithm of the drug release and the logarithm of time (F = F_max_ × 1 − e^[−((t − Ti)^β)/α]^) and does not have a kinetic and mechanistic nature. Nevertheless, it is possible to correlate the geometric form parameter β to the *n* exponent of semiempirical Korsmeyer–Peppas equation (F = k_KP_ × t*^n^*), which is used to describe the drug release from polymer matrices: it was shown by several authors that for β ≤ 0.75, Fickian diffusion is the release mechanism; for 0.75 < β < 1, anomalous transport involving both diffusion and swelling occurs; for β = 1, a non-Fickian case II drug release corresponding to zero-order kinetic regulated by swelling release mechanism is involved; and for β > 1, super case II transport occurs, in which a combination of diffusion, erosion and swelling is observed [38]. Since the average value of β = 0.295 ± 0.102 (Table 4), Fickian diffusion is the main driving force of the drug release in PR1_T small-diameter vascular grafts, thereby confirming that the solvent transport rate and diffusion is predominant over the relaxation of the polymeric matrix, making drug release time dependent [39]. Regarding PR2_T, even if a burst drug release could be observed in the first 9 h of drug release, the overall drug release rate was slower: 60% of the encapsulated tobramycin was released after 384 h (16 days) and drug release was not yet complete after 35 days of testing (Figure 6a). Kinetic modelling of the PR2_T drug release profile showed that Peppas–Sahlin was the best fitting model. The correlation factor R^2^ was 0.9909, indicating an excellent match between the predicted and real data. Moreover, an adjusted correlation parameter R^2-adj^ value of 0.9905, an AIC value of 100 and an MSC value of 4.51 indicated optimal model prediction power (Table 3). The Peppas–Sahlin model (F = k_1_ × t*^m^* + k_2_ × t^(2*m*)^) is derived from the Korsmeyer–Peppas equation: in addition to the constant k_1_ (equal to k_KP_), which describes the rate of drug release, and the exponent *m* (equal to *n*), which describes the release mode of the drug, the constant k_2_ is present in the equation, describing the rate of relaxation of the polymer chains and electrospun matrix [39]. Hence, in this case the drug release is regulated by two forces: *m* is equal to 0.45, indicating that tobramycin release occurs through Fickian diffusion, which is the main contributor since k_1_ = 5.233 ± 2.966; the second contribution is the relaxation of the matrix, for which the constant is k_2_ = −0.069 ± 0.149. The negative value of k_2_ indicates that the swelling of the polymer matrix had an impairing effect on drug release [40]. In fact, if we consider that Fickian diffusion is dependent from water diffusion into the polymer matrix, the slow hydration rate of PR2_T small-diameter vascular grafts perfectly matches with this drug release kinetic. 

Finally, the different drug release profiles and kinetics reported for PR1_T and PR2_T had a clear impact on microbicidal activity over *Staphylococcus aureus* and *Escherichia coli*. The two bacterial strains were selected because they were representative of vascular graft infections, covering both Gram-positive and Gram-negative microorganisms [4,8]. PR1_T and PR2_T graft samples were incubated in 10^7^ CFU/mL bacterial suspensions prepared in peptone water in order to avoid excessive starvation stress of the microorganism during the five days of testing. After 24 h the antibacterial effect was significant for both microorganisms and in the order of magnitude of a 1,000,000-fold reduction for *S. aureus* and a 100,000-fold reduction for *E. coli*. In PR1_T, the complete killing of the initial bacterial load was achieved after 72 h for *S. aureus* and after 48 h for *E. coli*. This strong antibacterial effect is due to the high burst effect of PR1_T that, after 24 h, is responsible for a tobramycin concentration of 11.02 ± 3.66 μg/mL and that, at the end of the 5 days of testing, reached 14.46 ± 4.36 μg/mL. In PR2_T the killing rate was slower, especially for *S. aureus,* in which a comparable 1,000,000-fold reduction in bacterial titre was reached after 48 h and kept constant until 72 h, when a further contraction of the bacterial population was observed. The initial bacterial population was completely broken down after 120 h. *E. coli* on the contrary showed a similar behaviour to PR1_T, with a 100,000-fold and a 100,000,000-fold reduction in the bacterial titre after 24 and 48 h, respectively. Tobramycin is an aminoglycoside with indication mainly against Gram-negative microorganisms, but due to its wide spectrum activity, it is also effective against certain Gram-positive bacteria, including *S. aureus* [14]. Nevertheless, *S. aureus* remains less susceptible to tobramycin explaining the lower bacterial killing rate. Moreover, PR2_T had a slower drug release rate that, after 24 h, gained a cumulative tobramycin concentration of 1.31 ± 0.79 μg/mL, reaching a concentration of 2.52 ± 2.04 μg/mL after 120 h. This means that comparing PR2_T to PR1_T, the latter exposed the tested microorganism to a 10-fold higher drug concentration, accelerating the bacteria killing rate. Therapeutic levels of tobramycin are generally considered between 4 and 6 μg/mL after the administration of a 1 mg/kg of body weight dose, via intravenous or intramuscular injection [41]. PR1_T widely covered the therapeutic levels of the drug, with the advantage to avoid systemic administration, whereas PR2_T had a good antimicrobial activity at lower doses, and both can be considered for localised drug delivery.

## 4. Materials and Methods

### 4.1. Materials

Poly(L-lactide-co-glycolide) (PLGA) (Resomer LG 824S, Lactide:Glycolide 82:18 ester terminated) was purchased from EVONIK ROHM GmbH (Essen, North Rhine-Westphalia, Germany). Dichloromethane (DCM); N, N-dimethylformamide (DMF); sodium metabisulphite (Na_2_S_2_O_5_); ethanol 96°; orthophosphoric acid 85%; and hydrochloric acid 37% were purchased from Carlo Erba Reagents (Cornaredo, Milan, Italy). Phenylacetaldehyde, citric acid, phosphate-buffered saline (PBS) tablets, tobramycin, ninhydrin were supplied from Sigma-Merck (Milan, Italy). Ultrapure water (0.067 μS/cm) was obtained from Q-POD equipped with 0.22 μm Millipore Corporation Millipak express 40 filter (Burlington, MA, USA). Sodium hydroxide pellets were purchased from AppliChem GmbH (Darmstadt, Hessen, Germany). The 25-gauge needle for electrospinning were supplied by Nordson EFD (West-lake, OH, USA). Tryptone soya broth (TSB) and tryptone soy agar (TSA) were supplied by Oxoid (Basingstoke, Hampshire, UK). Buffered peptone water and Eugonic broth (with lecithin, Triton X-100, polysorbate 80) were purchased from Thermo Fisher (Waltham, MA, USA). *Staphylococcus aureus* ATCC 6538 and *Escherichia coli* ATCC 10536 were supplied by LGC Standards (Sesto San Giovanni, Milan, Italy).

### 4.2. Methods

#### 4.2.1. Polymer Solutions Preparation and Characterization

PLGA solutions at 10% *w*/*v* and 12.5% *w*/*v* were prepared in 75:23.8:1.2 dichloromethane/N,N-dimethylformamide/0.01% *w*/*v* sodium metabisulfite aqueous solution (DCM/DMF/Na_2_S_2_O_5_) solvent system, as previously described [15]. Briefly, PLGA was allowed to completely dissolve in DCM for 8 h, then DMF and Na_2_S_2_O_5_ solution were added and stirred for at least 90 min to form a homogeneous solution. In drug-loaded polymer solutions, 0.2% *w*/*w_PLGA_* of tobramycin was first dissolved in the water phase, then reprecipitated in DMF and added to a PLGA/DCM solution to form a suspension. 

Rheological characterisation was performed using a Malvern Kinexus Pro+ rotational rheometer (NETZSCH-Gerätebau GmbH, Verona, Italy) equipped with a CP4/40 conical geometry (cone angle 1°, diameter 40 mm, gap from sample plate 0.15 mm) and a solvent trap. The linear viscoelastic region (LVER) was investigated by performing an amplitude sweep test from 0.01% to 1000% shear strain and at a constant frequency of 1 Hz. Dynamic viscosity was determined within the LVER by performing a shear rate ramp test, and zero-shear viscosity (Pa × s) was extrapolated as the intercept with the Y-axis of the straight line interpoling viscosity data points in the viscosity (Pa × s) vs. shear rate (s^−1^) bi-log plot [42]. 

Polymer solution conductivity measurements were performed using a Metrohm 914 conductometer (Metrohm Italia, Origgio, Varese, Italy). Both rheological and conductimetry analyses were performed at a temperature of 37 °C and 41 °C for PLGA 10% *w*/*v* and PLGA 12.5% *w*/*v* solutions, respectively.

Theoretical shear stress (τ_theo_) applied on both 10% *w*/*v* PLGA and 12.5% *w*/*v* PLGA polymeric solutions extruded through the needle during electrospinning was calculated according to Equation (1) [43]: τ_theo_ (Pa) = ^●^γ × η_0_(1)
where ^●^γ (s^−1^) is the shear rate of the polymer solution extruded through the needle, and η_0_ (Pa × s) is the zero-shear viscosity of the considered solution. ^●^γ was calculated by applying Equation (2) [44]:^●^γ (s^−1^) = 4Q/πr^3^(2)
where Q (mL/s) is the flow rate of the polymer solution during electrospinning, and r (mm) is the internal radius of the needle. 

#### 4.2.2. Vascular Graft Prototypes Electrospinning and Morphological Characterization

Vascular graft Prototype 1 (PR1) and Prototype 2 (PR2) were prepared by electrospinning the 12.5% and 10% *w*/*v* PLGA solutions, respectively. The polymer solution was loaded in a 5 mL syringe, extruded through a 25-gauge needle and electrospun with a NANON 01A vertical electrospinning device (MECC Co. Instruments Ltd., Fukuoka, Japan). Depending on the polymer concentration, different flow rates and voltages were set (see Table 1). The electrospun fibres were collected on a rotating 2.5 mm diameter rod to form a tubular fibrous scaffold. For the production of placebo vascular graft prototypes (PR1_PL and PR2_PL), plain PLGA solutions were electrospun, whereas to obtain tobramycin loaded vascular grafts (PR1_T and PR2_T), 0.2% *w*/*w_PLGA_* tobramycin-doped polymer solutions were used.

Vascular graft prototype wall thickness was measured with a digital calliper. Fibre and pore morphology was evaluated using Scanning Electron Microscopy (SEM): vascular graft prototype samples were fixed on microscopy stubs with carbon tape, gold coated by vapor deposition (11 mA, 120 s) and observed at different magnifications (1000×; 3000×; 30,000×) using a Zeiss EVO MA 10 scanning electron microscope (Carl Zeiss, Oberkochen, Baden-Württemberg, Germany). SEM images were analysed using ImageJ 1.54f software (Wayne Rasband and contributors, National Institutes of Health, USA) [45]: the line tool was used to measure fibre diameter and the polygon tool was used to outline the pores’ perimeter and calculate the Ferret’s diameter. NiBlack thresholding was used to obtain local threshold images for overall percentage porosity analysis.

#### 4.2.3. Water up Take Assay

Hydration profiles of tobramycin-loaded vascular graft prototypes were investigated by submerging 5 cm long graft samples (n = 3) in 15 mL of PBS at pH = 7.4 and incubated at 37 °C. At fixed time points (9, 24, 48, 72, 144, 168, 192 and 216 h), the samples were removed from the medium, carefully blotted on tissue paper and weighed. Water uptake percentage was calculated according to Equation (3): WU (%) = (m(*t*) − m*i*/m*i*) × 100(3)
where m(*t*) is the weight (mg) of the graft at the sampling point after blotting and m*i* is the initial weight of the dry graft. 

#### 4.2.4. Drug Encapsulation Efficiency, Vascular Grafts Drug Content and In Vitro Drug Release Evaluation

The tobramycin encapsulation efficiency in electrospun fibres was evaluated by extracting the drug from vascular graft prototypes: whole drug-loaded vascular grafts were cut in small fragments and dissolved in 8 mL of DCM and added to 10 mL of ultrapure water. The biphasic system was stirred at 600 rpm for 1 h, and the aqueous supernatant collected to quantify the amount of extracted tobramycin. Extraction efficiency was assessed by performing extraction assays using placebo vascular grafts and by adding known amounts of tobramycin to the organic phase prior to extraction. The encapsulation efficiency (EE%) and overall drug content (DC) were determined on five PR1_T and five PR2_T grafts by applying the following Equations (4) and (5):EE% = ExT/TheoT × 100(4)
where ExT is the extracted tobramycin from the graft (μg), and TheoT is the theoretical amount (μg) of 100% encapsulated tobramycin based on the graft weight.
DC (μg/mg) = ExT/m(5)
where ExT is, again, the extracted tobramycin from the graft (μg), and m is the weight of the graft.

In vitro drug release assays were performed by immersing whole PR1_T and PR2_T (n = 5) in 6 mL of PBS at pH = 7.4, supplemented with 0.01% *w*/*v* Na_2_S_2_O_5_ and incubated at 37 °C in static and sink conditions. At fixed time points (0, 3, 6, 9, 24, 48, 72, 96, 168, 192, 216, 240, 264, 336, 384, 432, 504, 552, 600, 720 and 840 h), 3 mL of release medium was sampled and stored at −20 °C until further quantification. The total medium volume was reconstituted with 3 mL of fresh PBS. Moreover, an in vitro drug dissolution assay was performed: 60 μL of a 1 mg/mL tobramycin stock solution prepared in methanol was deposited on the bottom of a well of a 6-well plate; the tobramycin solution drop was allowed to evaporate at 37 °C for 40 min, leaving 60 μg of tobramycin residue. 3 mL of PBS were added extremely slowly to the well to limit turbulent flow, and the 6-well plate was incubated at 37 °C in static conditions. At fixed time points of 0, 3, 6, 9, 24, and 48 h, 1 mL of dissolution media was sampled and stored at −20 °C until further quantification. The total medium volume was reconstituted with 1 mL of fresh PBS.

Tobramycin quantification was performed by adapting a method reported in the literature [46]. Tobramycin’s primary ammine group was reacted with ninhydrin and phenylacetaldehyde to obtain a fluorescent product analysed using a Shimadzu RF-6000 RP spectrofluorometer (Shimadzu Italia, Milan, Italy). 1 mL of sample containing tobramycin was added to 1 mL of Teorell–Stenhagen buffer at pH = 6.0 ± 0.1 and reacted with 0.6 mL of a 5.6 mM ninhydrin aqueous solution and 0.6 mL of a 1.8 mM phenylacetaldehyde solution in ethanol, and incubated at 75 °C for 10 min. The reaction was quenched in ice water, and the solution was analysed at λ_exitation_ = 395 nm and λ_emission_ = 470 nm. PBS samples were collected at the same sampling time from PBS-immersed placebo grafts and reacted as previously mentioned, were used as blank. To relate the average fluorescence intensity (n = 3) to tobramycin concentration, a calibration curve was built by preparing tobramycin standard solutions with concentration ranging from 30 to 0.312 μg/mL in PBS. The standards were reacted as previously mentioned and further diluted 1:10 with ethanol before fluorescence analysis (n = 3), using PBS again as blank. The final calibration curve ranged from 3 to 0.03 μg/mL and had a correlation coefficient of R^2^= 0.9989.

#### 4.2.5. Drug Release Kinetic Study

Release profile data sets of both PR1_T and PR2_T underwent non-linear least-squares curve-fitting to mathematical drug release kinetic models, using open access Microsoft Excel add-in DDSolver [37]. Cumulative percentage drug release data (n = 5) were entered into an Excel spreadsheet according to the template supplied by the developers and the fitting to 7 different equations was performed, including Higuchi, Korsmeyer–Peppas, Peppas–Sahlin, Weibull, Logistic, Gompertz and Probit equations. The goodness of fit of the different models were estimated through different statistical criteria provided by the program. Specifically, the correlation factor (R^2^), corrected correlation factor (R_2-adj_), Akaike Information Criterion (AIC) and Model Selection Criterion (MSC) were used to discriminate the most appropriate model for each prototype.

#### 4.2.6. Antimicrobic Assay

Assessment of antimicrobic efficacy of PR1_T and PR2_T was performed as previously described by the authors [15]. Briefly, *Staphylococcus aureus* ATCC 6538 and *Escherichia coli* ATCC 10536 were cultured overnight at 37 °C in tryptone soya broth. The obtained cultures were centrifuged (2000 rpm for 25 min), the supernatant was removed, and the microorganism pellet was resuspended in sterile water to reach a bacterial titre of 10^7^ CFU/mL. The final testing bacterial suspensions were prepared by diluting 1 mL of each microbial suspension in 4mL of peptone water. PR1_T and PR2_T grafts were cut into 3 cm long pieces and sanitised under UV light overnight. Each sample was immersed in a testing bacterial suspension and incubated at 37 °C in aerobic conditions. Placebo PR1_PL and PR2_PL grafts were used as positive bacterial growth control. At fixed times (24, 48, 72, 96 and 120 h), a 50 μL aliquot of the bacterial suspension was collected and serially diluted in an antibiotic neutralising agent (Eugonic broth). 1 mL of each dilution was inoculated into tryptone soya agar (TSA), and the plates were incubated for 24 h at 37 °C. After incubation, viable bacterial titres were calculated. The microbicidal effect (ME) at each time of the tobramycin-loaded grafts was calculated using Equation (6):ME = log_10_ [PRn_PL] − log_10_ [PRn_T](6)
where [PRn_PL] is the viable bacterial titre in CFU/mL after contact with each placebo prototype graft, and [PRn_T] is the viable bacterial titre in CFU/mL after contact with the prototype of the tobramycin-loaded graft.

#### 4.2.7. Statistical Analyses

GRAPHPAD Prism 9.0 (San Diego, CA, USA) was used for generating all graphs and for statistical analysis. 2-sample *t*-tests were performed to determine the significance between the two experimental groups. Statistical significance is represented as * for *p* < 0.05, ** for *p* < 0.01, *** for *p* < 0.005, **** for *p* < 0.001 and ns (*p* > 0.05) for non-significant.

## 5. Conclusions

In this work, the influence of the primary and secondary morphology of nanostructured small-diameter vascular grafts on hydration and drug release kinetics was highlighted. Peculiar secondary fibre morphology can be obtained by optimising the combination of solution and electrospinning process parameters, and an enhanced but still prolonged local antibiotic release can be obtained by introducing porosity onto the fibres surface. On the contrary, the absence of such porosity ensured a more extended drug release up to 35+ days. A good in vitro antibacterial activity was obtained for both Gram-positive- and Gram-negative-representative bacterial strains, even by lowering the local tobramycin concentration from about 15 to 3 μg/mL. Nevertheless, to further investigate in vitro drug release and antimicrobic efficiency, a more complex and implantation site-mimicking experimental setup will be designed and tested. Moreover, the combination of different types of fibre morphology in a single small-diameter vascular graft is desirable in order to merge different drug release kinetics and mechanisms so as to cover both early and delayed vascular graft infections. Future steps will deepen the aspects related to interactions between grafts and a biologic environment. They will involve haemocompatibility tests and testing the grafts in simulated body fluids. Moreover, the addition of ECM-like molecular cues will be evaluated to promote the attachment, differentiation and proliferation of relevant cell lines on the small-diameter vascular graft prototypes and evaluate not only the scaffolds’ suitability for vascular regeneration but also the influence of cell ingrowth on the mechanical properties of the grafts.

## Figures and Tables

**Figure 1 ijms-24-12108-f001:**
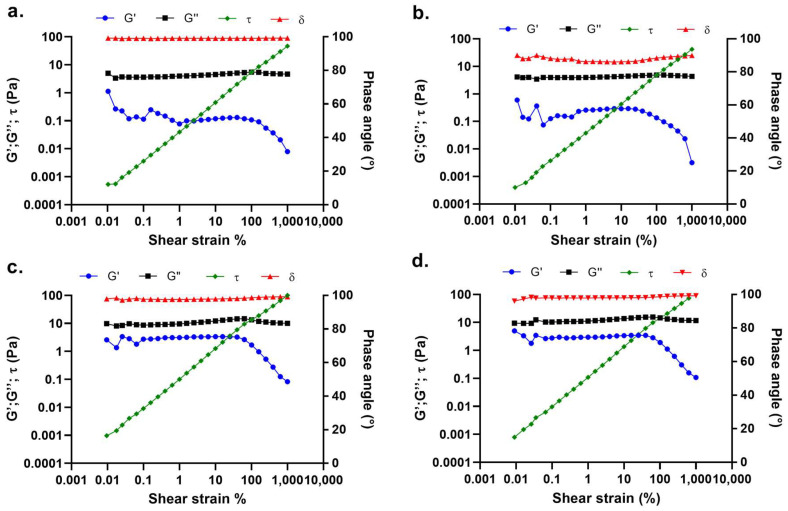
Amplitude sweep tests performed on (**a**) 10% *w*/*v* PLGA placebo solution, (**b**) 10% *w*/*v* PLGA 0.2% *w*/*w_PLGA_* tobramycin-doped polymer solution, (**c**) 12.5% *w*/*v* PLGA placebo solution, (**d**) 12.5% *w*/*v* PLGA 0.2% *w*/*w_PLGA_* tobramycin-doped polymer solution. Elastic modulus (G′), viscous modulus (G″), shear stress (τ) and phase angle (δ) are plotted versus shear strain.

**Figure 2 ijms-24-12108-f002:**
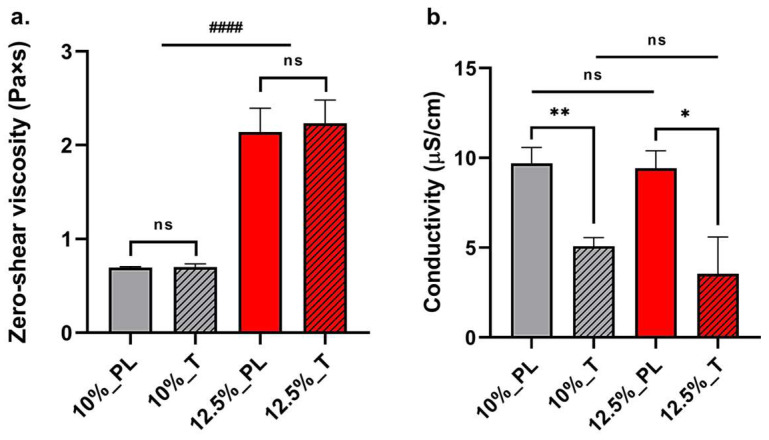
(**a**) Average zero-shear viscosity ± standard deviation (n = 3) and (**b**) average conductivity ± standard deviation (n = 3) of 10% *w*/*v* PLGA placebo solution (10%_PL), 10% *w*/*v* PLGA 0.2% *w*/*w_PLGA_* tobramycin-doped polymer solution (10%_T), 12.5% *w*/*v* PLGA placebo solution (12.5%_PL), 12.5% *w*/*v* PLGA 0.2% *w*/*w_PLGA_* tobramycin-doped polymer solution (12.5%_T). Statistical analysis was performed between placebo and tobramycin-doped polymer solutions with same concentration and between 10% and 12.5% *w*/*v* PLGA concentration groups (parametric *t*-test between PL and T: statistical significance is represented as * for *p* < 0.05, ** for *p* < 0.01, otherwise non-statistical significant (ns); parametric *t*-test between 10% and 12.5% *w*/*v* PLGA concentration groups: statistical significance is represented as #### for *p* < 0.001).

**Figure 3 ijms-24-12108-f003:**
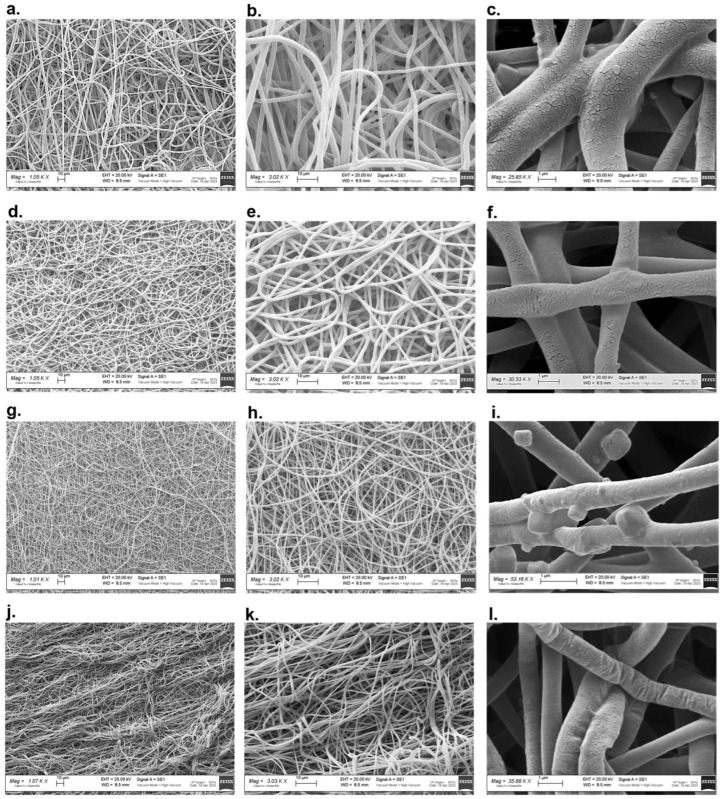
Scanning electron microscopy images taken at different magnifications (1000×, 3000× and 25,000–35,000×) of (**a**–**c**) Prototype 1 placebo; (**d**–**f**) Prototype 1 tobramycin loaded; (**g**,**h**) Prototype 2 placebo; (**i**) Prototype 2 placebo with PBS crystals; (**j**–**l**) Placebo 2 tobramycin loaded.

**Figure 4 ijms-24-12108-f004:**
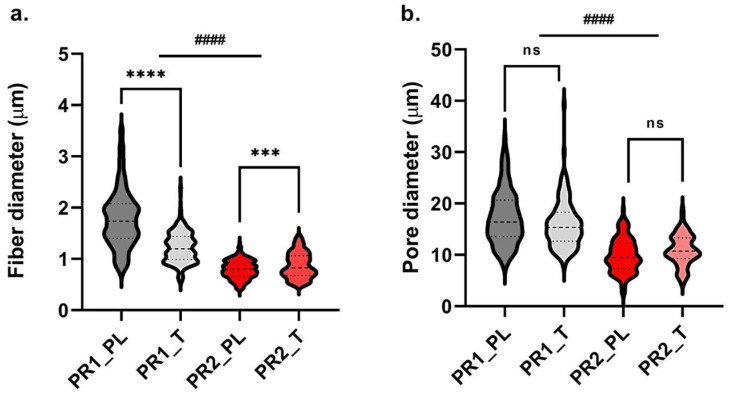
Fibre (**a**) and pore (**b**) diameter distributions of PR1_PL and PR1_T, and PR2_PL and PR2_T. Central dashed line is median, lower dotted line is first quartile, upper dotted line is third quartile (for fibre diameter n = 200; for pore size n = 180). Statistical analysis was performed between placebo and tobramycin-doped grafts of the same prototype, and between PR1 and PR2 graft groups if non-statistical differences were detected among placebo and tobramycin-loaded grafts (parametric *t*-test between PL and T: statistical significance is represented as *** for *p* < 0.005, **** for *p* < 0.001, otherwise non-statistical significant (ns); parametric *t*-test between PR1 and PR2 groups: statistical significance is represented as #### for *p* < 0.001).

**Figure 5 ijms-24-12108-f005:**
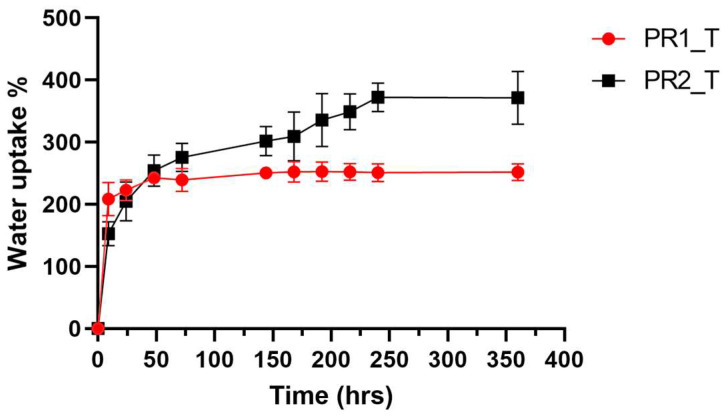
Water uptake percentage plotted against time of PR1_T and PR2_T, submerged in PBS pH = 7.4 and incubate at 37 °C. Data presented as average ± standard deviation (n = 3).

**Figure 6 ijms-24-12108-f006:**
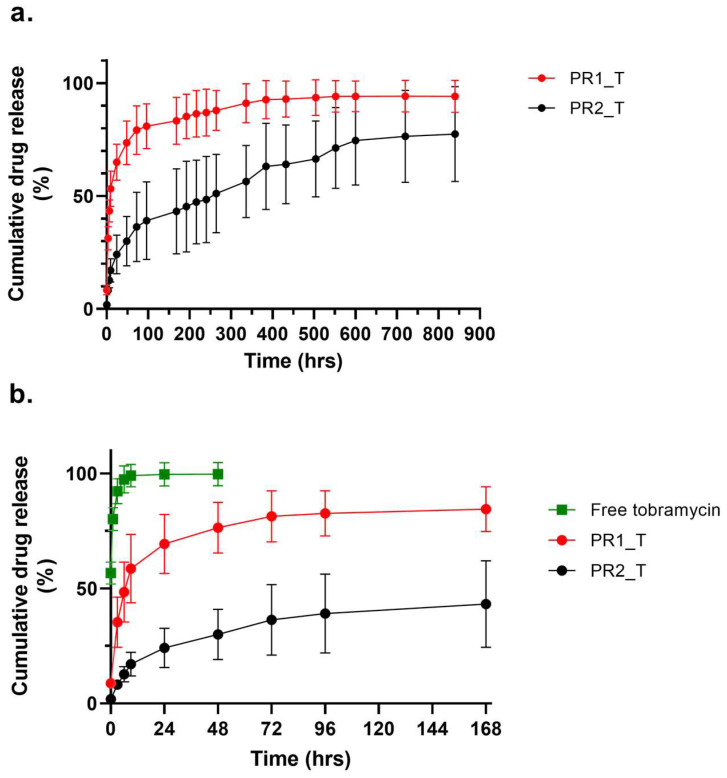
Cumulative tobramycin release profiles of (**a**) PR1_T and PR2_T during the whole in vitro drug release experiment; (**b**) plot extract of first week of drug release of PR1_T and PR2_T, compared to dissolution profile of free tobramycin. Data presented as average ± standard deviation (n = 5).

**Figure 7 ijms-24-12108-f007:**
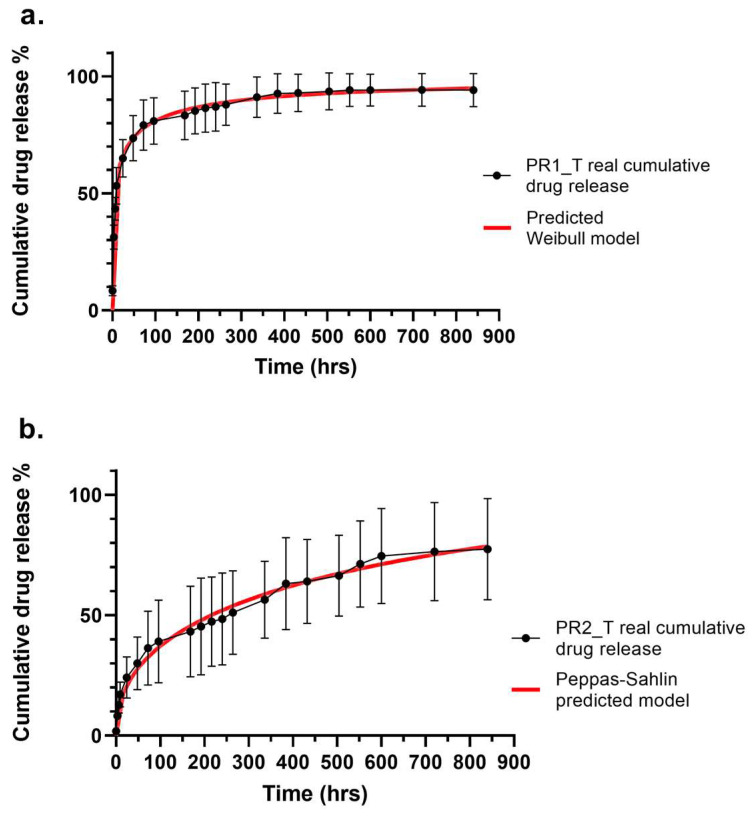
Superimposed real (black) and predicted (red) cumulative drug release profiles of (**a**) Prototype 1 and (**b**) Prototype 2. Real drug release data were reported as average ± standard deviation (n = 5); predicted drug release data reported as average (n = 5). Standard deviation was equal to 8% for Weibull predicted data, and equal to 20% for Peppas–Sahlin predicted data.

**Table 1 ijms-24-12108-t001:** Electrospinning parameters applied for the production of vascular graft prototypes. Prototype 1 was manufactured using a 12.5% *w*/*v* polymer solution, whereas Prototype 2 was obtained from a 10% *w*/*v* polymer solution. The same electrospinning parameters were applied to produce both placebo and tobramycin-loaded vascular grafts.

PLGA Concentration (% *w*/*v*)	Voltage (kV)	Flow Rate (mL/h)	Temperature Range (°C)	Relative Humidity (%)
10	18	0.1	30–35	20
12.5	20	0.3	45–37	20

**Table 2 ijms-24-12108-t002:** Grafts’ mean wall thickness, mean fibre and pore diameter, and average overall percentage porosity measurements for PR1_PL and PR1_T grafts, and PR2_PL and PR2_T grafts. Data reported as average ± standard deviation (for graft wall thickness n = 3; for fibre diameter n = 200; for pore diameter n = 180; for overall percentage porosity n = 3).

Graft Type	Wall Thickness (μm)	Fibre Diameter (μm)	Pore Diameter (μm)	Porosity (%)
PR1_PL	291.95 ± 32.64	1.784 ± 0.561	17.20 ± 5.24	43.79 ± 0.90
PR1_T	291.00 ± 20.82	1.224 ± 0.301	16.00 ± 4.691	45.82 ± 0.44
PR2_PL	180.00 ± 18.00	0.794 ± 0.178	9.91 ± 3.27	44.00 ± 2.64
PR2_T	167.10 ± 13.08	0.870 ± 0.248	10.96 ± 3.00	44.49 ± 3.77

**Table 3 ijms-24-12108-t003:** Goodness of fit of each mathematical model with Prototype 1 and Prototype 2 release data was performed by calculating correlation factor (R^2^) and by comparing the corrected correlation factor (R^2-adj^), Akaike information criterion (AIC) and model selection criterion (MSC) calculated for different models within the same prototype. Statistical parameters were calculated using average drug release data (n = 5).

Mathematical Model	Prototype 1				Prototype 2			
	R^2^	R^2-adj^	AIC	MSC	R^2^	R^2-adj^	AIC	MSC
Higuchi	−0.2953	−0.2953	203	−0.3540	0.9386	0.9386	138	2.6959
Korsmeyer–Peppas	0.9332	0.9297	143	2.5158	0.9886	0.9874	107	4.1912
Peppas–Sahlin	0.8304	0.8116	164	1.4887	0.9909	0.9905	100	4.5131
Weibull	0.9978	0.9891	108	4.1387	0.9643	0.9580	133	2.9507
Logistic	0.9887	0.9874	109	4.1947	0.9276	0.9195	146	2.3396
Gompertz	0.9834	0.9816	116	3.8131	0.8914	0.8794	154	1.9348
Probit	0.9899	0.9888	105	4.4096	0.9356	0.9285	143	2.4576

**Table 4 ijms-24-12108-t004:** Best fitting mathematical models describing drug release kinetic of Prototype 1 and Prototype 2 vascular grafts, corresponding model equation and parameter values. Data were reported as average ± standard deviation (n = 5).

Prototype	Best Fitting Model	Equation	Parameters
PR1	Weibull	F = F_max_ × (1 − e^[−((t − Ti)^β)/α]^)	α = 2.557 ± 1.250 β = 0.295 ± 0.102 Ti = 1.500 ± 0.329
PR2	Peppas–Sahlin	F = k_1_ × t*^m^* + k_2_ × t^(2*m*)^	k_1_ = 5.233 ± 2.966 k_2_ = −0.069 ± 0.149 *m* = 0.45

**Table 5 ijms-24-12108-t005:** Microbicidal effect (ME) of PR2_T graft on *S. aureus* and *E. coli* for up to of 5 days. Data were reported as average ± standard deviation (n = 3).

Time (h)	ME	ME
	*S. aureus*	*E. coli*
24	3.98 ± 1.47	5.89 ± 1.16
48	6.31 ± 0.41	8.82 ± 0.07
72	6.30 ± 0.15	8.42 ± 0.40
96	7.55 ± 1.50	8.15 ± 0.53
120	8.58 ± 0.05	8.39 ± 0.31

## Data Availability

Not applicable.

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
