# Peer review of "Influence of Electrospun Fibre Secondary Morphology on Antibiotic Release Kinetic and Its Impact on Antimicrobic Efficacy"

_ijms, 2023, doi:10.3390/ijms241512108_

Round 1
Reviewer 1 Report
In this paper, the authors explored the impact of the electrospun fiber secondary morphology on antimicrobic efficacy. Since the methods and results look appropriate, I believe the manuscript has good perspective but is currently in the draft. Serious work is required.
1. The Introduction section does not provide context to help readers understand the significance of your research. We suggest you consider adding more detail to help readers better understand the importance of your work.
2. In Figure 6, we recommend verifying the cumulative tobramycin release profiles in other solutions, such as tissue fluid.
3. The manuscript does not provide any direction on future research or the next steps following this study's completion. Please consider providing some guidance in this area.
It is noted that your manuscript needs careful editing by someone with expertise in technical English editing paying particular attention to English grammar, spelling, and sentence structure so that the goals and results of the study are clear to the reader.
Author Response
For the the answers to the Reviewer's comments please see attachment.

Reviewer 2 Report
Dear Authors,
The experimental data from your manuscript "Influence of electrospun fibre secondary morphology on antibiotic release kinetic and its impact on antimicrobic efficacy" are of great scientific and medical interest.
The introductory part draws a pertinent overview of the current research and medical situation.
The methodology is well adapted to the study, and well presented.
However, the fact that the presentation of the experimental results is separated from the discussion part makes this manuscript incredibly difficult to read and fully appreciate. The very long and very descriptive results presentation part, with long sentences full of dense information and without any interpretation of the data, is not the best way to report valuable scientific information, at least for this very specific manuscript.
Citing and using some of your own experimental data already reported, to highlight new original aspects in relation to the data in this study, is a coherent bonus in my view.
In conclusion, I consider the literature data, methodology and experimental results to be of excellent quality. However, I recommend a major revision of the current version, to merge the results and discussion parts, in order to allow the reader a more relevant and coherent presentation and to increase the readability and scientific logic of the entire work.
Author Response
For the answers to the Reviewer's comments please see attachment

Round 2
Reviewer 2 Report
I recommend this version for publication, as it is.